



# Massive simplification of the wind farm layout optimization problem

**Andrew P. J. Stanley and Andrew Ning**

Department of Mechanical Engineering, Brigham Young University,
701 E University Pkwy, 350 EB, Provo, UT 84602, USA

**Correspondence:** Andrew P. J. Stanley (stanley_andrewpj@byu.edu)

**Abstract.** The wind farm layout optimization problem is notoriously difficult to solve because of the large number of design variables and extreme multimodality of the design space. Because of the multimodality of the space and the often discontinuous models used in wind farm modeling, the wind industry is heavily dependent on gradient-free techniques for wind farm layout optimization. Unfortunately, the computational expense required with these methods scales poorly with increasing numbers of variables. Thus, many companies and researchers have been limited in the size of wind farms they can optimize. To solve these issues, we present the boundary-grid parameterization. This parameterization uses only five variables to define the layout of a wind farm with any number of turbines. For a 100-turbine wind farm, we show that optimizing the five variables of the boundary-grid method produces wind farms that perform just as well as farms where the location of each turbine is optimized individually, which requires 200 design variables. Our presented method facilitates the study and both gradient-free and gradient-based optimization of large wind farms, something that has traditionally been less scalable with increasing numbers of design variables.

## 1 Introduction

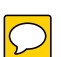

In 2018, wind energy produced 6.6 % TS1 of the electricity use in the United States.[1] With current market trends and technology, the U.S. Energy Information Administration projects that this number will rise by 1 % in both 2019 and 2020 (U.S. Energy Information Administration, 2019a), and the installed capacity will increase by 4 % every year through 2050 (U.S. Energy Information Administration, 2019b). In order for the US and the rest of the world to meet and exceed these projections, it is necessary to be able to create efficient turbine layouts for large wind farms. The wind farm layout optimization problem is notoriously difficult to solve because of the large number of design variables, computationally expensive models for high fidelity simulations, and extreme multimodality of the design space (see Fig. 1).

Because of the multimodality of the space and the often discontinuous models used in wind farm modeling, the wind industry is heavily dependent on gradient-free techniques for wind farm layout optimization (Herbert-Acero et al., 2014). Although these methods can be highly effective for small numbers of design variables, the computational expense required to converge scales poorly, approximately quadratically, with increasing numbers of variables (Singg et al., 2008; Rios and Sahinidis, 2013; Lyu et al., 2014; Ning and Petch, 2016; Thomas and Ning, 2018). Because of this poor computational scaling, many companies and researchers have been limited in the size of wind farms they can optimize, as the number of variables typically increases with the number of turbines. Figure 2 demonstrates this principle. This figure shows the number of function evaluations required to optimize the multi-dimensional Rosenbrock function versus the number of variables (Rosenbrock, 1960). To give a sense of what these numbers mean, if this problem with 64 variables and exact-analytic gradients takes 1 h to

[1] https://www.eia.gov/tools/faqs/faq.php?id=427&t=3 (last access: 9 December 2019).

(a) 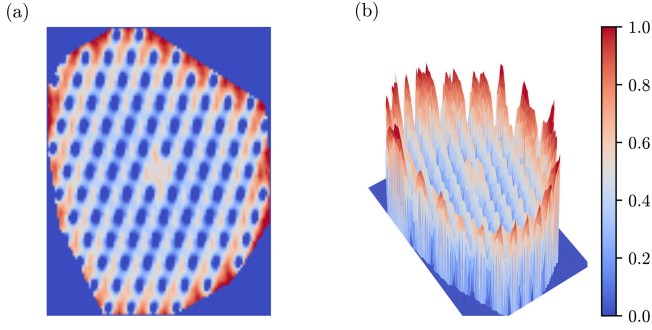 (b)

**Figure 1.** The complexity and multimodality of wind farm layout design space. Shown is the normalized annual energy production of a 100-turbine wind farm as a function of the location of one turbine; 99 turbines remain fixed, while one is moved throughout the wind farm. **(a)** A 2-D view of the design space. **(b)** A 3-D surface, which highlights the extreme variation of the peaks and valleys. This figure shows only the multimodality from two dimensions, where the true design space has 200 design variables.

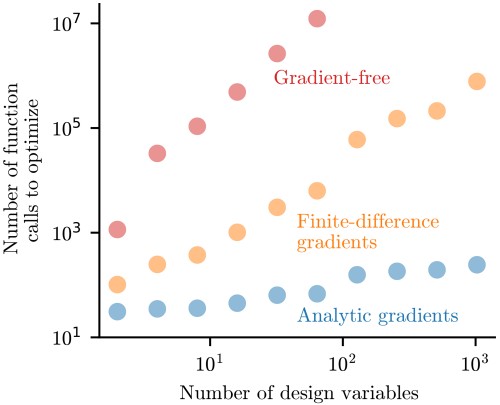

**Figure 2.** The number of function calls required to optimize the multi-dimensional Rosenbrock function versus the number of variables. The computational expense of gradient-free and finite-difference gradients scale poorly with the number of variables.

optimize, using finite-difference gradients would take almost 4 d, while a gradient-free method would take over 20 years. The trends, not the exact numbers, shown in this figure are general for other optimization problems, such as wind farm layout. As the size of the problem increases, the computational expense with certain optimization methods can become unmanageable.

Despite its difficulty, layout optimization is an essential step in wind farm development in order to maximize power production. Power losses of 10 %–20 % are typical from turbine interactions within a wind farm (Barthelmie et al., 2007, 2009; Briggs, 2013) and can be as high as 30 %–40 % for farms with turbines spaced within 3 rotor diameters of each other (Stanley et al., 2019). However, because the difficulties in finding optimal turbine placement increase with the number of turbines, layout optimization can quickly become

infeasible for large wind farms (Ning and Petch, 2016). Even so, accelerated research and understanding of the principles governing wind energy as well as public demand for renewable energy sources are encouraging developers and communities to install farms with more wind turbines than have been typical in the past. Current turbine layout definitions and optimization methods are woefully inadequate for these increasingly large farms.

The most common current wind farm layout definitions include defining the location of each turbine directly (Feng and Shen, 2015; Guirguis et al., 2016; Gebraad et al., 2017), pre-assigning some locations in a wind farm as suitable turbine locations to limit the size of the design space (Emami and Noghreh, 2010; Parada et al., 2017; Ju and Liu, 2019) and parameterizing the turbines as a grid (Neubert et al., 2010; González et al., 2017; Perez-Moreno et al., 2018). Defining the location of every wind turbine directly allows the most freedom but also requires two variables for each turbine. In addition, the design space is the most multimodal. If one limits the design space by predetermining acceptable turbine locations or parameterizing the turbine locations with a simple grid, they are able to optimize larger wind farms. However, these methods produce simplistic wind farm designs, which underperform for most realistic scenarios.

In this paper we present the boundary-grid (BG) layout parameterization, a new wind farm layout parameterization. This new method solves the challenges that have previously made wind farm layout optimization so difficult. BG parameterization uses only five variables and can produce layouts that perform just as well as or better than the layouts achieved by directly optimizing the location of each wind turbine. With some of the most advanced wind farm optimization methods that have previously been available, we can directly optimize the location of every turbine in a 100-turbine wind farm in 4–5 h. More common methods take on the order of days or longer. With BG parameterization, we can optimize a 100-turbine wind farm in 3 min. Additionally, this new parameterization dramatically reduces the multimodality of the design space compared to direct layout optimization (compare Figs. 1 and 13b). Finally, BG parameterization has additional benefits, including a regular, aesthetically pleasing layout and naturally defined roads or shipping lanes. This technique can immediately be applied to wind farm design to obtain excellent wind farm layouts with limited computational resources.

## 2   Boundary-grid parameterization

When the locations of wind turbines in a farm are optimized directly, the final layout often follows two general rules. First, a large fraction of turbines are grouped on or near the wind farm boundary. Second, the turbines that are not positioned on the boundary are loosely arranged in rows throughout the farm (Fig. 3a). By observing these patterns in optimal

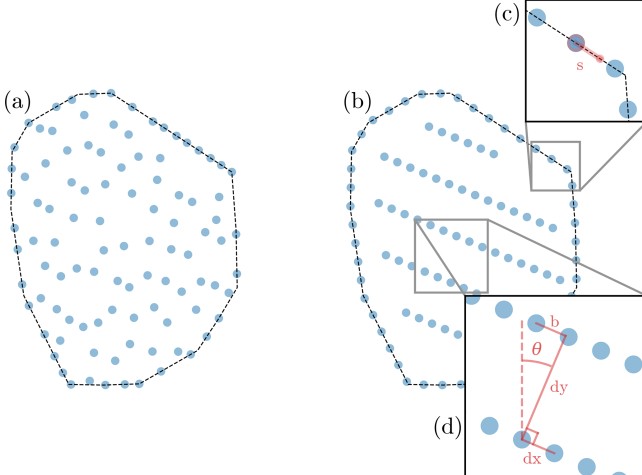

**Figure 3.** Example 100-turbine wind farm layouts, and parameterized wind turbine layout definition. Each dot is to scale, representing the wind turbine diameter. **(a)** Wind farm layout when the position of each turbine has been optimized directly. This optimization required 200 design variables – the $x$ and $y$ location of each turbine. **(b)** Wind farm layout optimized with boundary-grid parameterization. This optimization required five design variables, shown in panels **(c)** and **(d)**. **(c)** The start location design variable, $s$. **(d)** The four variables defining the inner grid: the grid spacing, $dx$ and $dy$, the grid offset $b$, and the rotation, $\theta$.

wind farm layouts, we defined our new layout parameterization such that it would create wind farms that filled these requirements.

## 2.1 New layout variables

In BG parameterization, the turbines are divided into two groups: the boundary and the inner grid (Fig. 3b). The boundary turbines are spaced around the circumference of the wind farm and are defined with one design variable. The rest of the turbines in the farm make up the inner grid, which is defined with four design variables for a total of five variables to describe the location of every turbine in the farm. The boundary turbines are placed on the wind farm boundary, spaced equally traversing the perimeter. These are defined by one variable, $s$, which is the distance along the perimeter where the first turbine, or start turbine, is placed. This in turn defines the position of every turbine around the boundary (Fig. 3c). During the development of our parameterization method, we tested various strategies of spacing the turbines around the boundary. However, we found that equally spacing the turbines around the perimeter consistently provided the best results. The inner grid turbines are defined by four design variables: $dx$, $dy$, $b$, and $\theta$. The grid spacing, $dx$ and $dy$, is the distance between columns and rows in the grid; $b$ is the offset distance, which defines how far consecutive rows are offset; $\theta$ is the grid rotation angle, which rotates the entire grid (Fig. 3d). The grid offset could also be defined as an angle;

however, we have used a distance as the gradients are more conducive to optimization. The inner grid is centered around the wind farm center, ensuring a one-to-one mapping from the design variables to the possible wind farm layouts.

## 2.2 Selection of discrete values

There are some discrete values which are important in our formulation, namely the number of turbines which are placed along the boundary and how many are in the grid, how many rows and columns are in the grid, and how the rows and columns are organized. We present some rules that we have found effective in determining these discrete values for all wind roses, wind farm boundaries, and wake models that we tested. Each individual case may benefit slightly from a more specialized selection of these values but our method works well across all cases tested.

The number of turbines placed on the boundary is determined by the wind farm perimeter and turbine rotor diameter. If the perimeter is large enough, 45 % of the wind turbines are placed on the boundary. In some cases, the wind farm perimeter is small and would result in turbines that are too closely spaced if 45 % were placed around the boundary. In this case, the number of boundary turbines is reduced until the minimum desired turbine spacing in the wind farm is preserved. When defining the number of turbines to be placed along the perimeter, the user must consider the most extreme boundary angles, such that minimum turbine spacing is preserved even at boundary corners. No matter how many turbines are placed around the boundary, they are always spaced equally traversing the perimeter, and all of the remaining turbines are placed in the inner grid. Note that the number of boundary turbines is determined before the number of turbines in the inner grid, to ensure that sufficient spacing is maintained between the boundary turbines.

The number of rows, columns, and their organization in the grid is determined with the following procedure. First, $dy$ is set to be 4 times $dx$, $b$ is set such that turbines are offset 20° from those in adjacent rows, and $\theta$ is initialized randomly. Then, $dx$ is varied with $\theta$ remaining constant, and $dy$ and $b$ change to fulfill the requirements prescribed in the initialization definition, until the correct number of turbines are within the wind farm boundary. During optimization, each of the grid variables can change individually; however, the discrete values remain fixed. For extremely small wind farms, with an average turbine spacing much less than 4 rotor diameters, it may be impossible to initialize the turbine rows with $dy$ equal to 4 times $dx$ and meet the minimum spacing constraints. In this case, the discrete row variable initialization would need to be adjusted.

The process outlined to select the discrete variables used in the parameterization is recommended as a starting point, and when computational resources or time is limited. We tested many different methods of how to determine the discrete values, but found that the method shown above consistently pro-

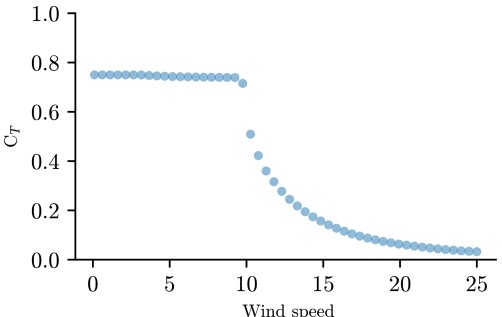

**Figure 4.** The thrust coefficient curve for the 3.35 MW turbine used in this paper.

duced wind farm layouts with high energy production. With sufficient resources, some scenarios may benefit from optimizing with a different ratio of boundary turbines or different initializations of the boundary grid. However, the results discussed in this paper were produced with the method given in this section. Because these variables are discrete, they cannot be included as design variables when using a gradient-based optimization method because the function space would be discontinuous. But a gradient-free optimization may benefit from including some of these discrete variables as design variables in the optimizations.

## 3   Wind farm modeling

### 3.1   Wind turbine parameters

In the testing of the BG wind farm layout parameterization method, we modeled the turbine parameters after the IEA 3.35 MW reference turbine (Bortolotti et al., 2019). The relevant parameters are a rotor diameter of 130 m, a hub height of 110 m, a rated aerodynamic power of 3.6 MW, and a generator efficiency of 93 %. The thrust coefficient curve for this turbine is shown in Fig. 4, and was generated using CCBlade, a blade element momentum code (Ning, 2013). The power curve was defined as a piecewise equation in Eq. (1).

$$
P_i(V) = \begin{cases} 0 & V < V_{\text{cut-in}} \\ P_{\text{rated}}\left(\dfrac{V}{V_{\text{rated}}}\right)^3 & V_{\text{cut-in}} \leq V < V_{\text{rated}} \\ P_{\text{rated}} & V_{\text{rated}} \leq V < V_{\text{cut-out}} \\ 0 & V \geq V_{\text{cut-out}} \end{cases} \tag{1}
$$

In this power curve definition, $P_i$ is the aerodynamic power produced by an individual wind turbine, $V$ is the hub velocity at that turbine (Lackner and Elkinton, 2007; Chen et al., 2015; Park and Law, 2015), $P_{\text{rated}}$ is 3.6 MW, $V_{\text{rated}}$ is 10 m s$^{-1}$, $V_{\text{cut-in}}$ is 3 m s$^{-1}$, and $V_{\text{cut-out}}$ is 25 m s$^{-1}$. The aerodynamic power is then multiplied by the generator efficiency to calculate the electric power.

### 3.2   Wind farm details

The major benefit of wind turbine layout parameterization comes for large wind farms. For farms with just a few turbines, the layout can be optimized directly with a small amount of design variables. In such cases with few design variables, there is little to no benefit gained from intelligently parameterizing the design space. In this study, each wind farm layout that we optimized had 100 wind turbines, to demonstrate the benefits of BG parameterization for large wind farms.

We tested the performance of our parameterization method on wind farms with different average turbine spacing: 4, 6, and 8 rotor diameters shown in Fig. 10. In addition to testing wind farms with different turbine spacing, we modeled and optimized several different wind farm boundaries in this study: the boundary of the Princess Amalia wind farm, a real farm in the North Sea (Van Dam et al., 2012; Gebraad and Van Wingerden, 2015; Kanev et al., 2018), a circle, and a square to demonstrate the sharp angles that can occur in wind farm boundaries. These boundaries are shown in Fig. 12.

### 3.3   Wake model

Wind speed deficits in this paper were predicted from turbine wakes with a modified version of the 2016 Bastankhah Gaussian wake model (Bastankhah and Porté-Agel, 2016). The original formulation of the model does not define the wake deficit in the near-wake region, creating undefined regions which make optimization difficult. To mitigate this issue, Thomas and Ning added a linear interpolation of the wake loss from the turbine up to where it is defined by the wake model, which is the version used in this paper (Thomas and Ning, 2018). The most important equation for this Gaussian wake model is shown in Eq. (2):

$$
\frac{\Delta \overline{u}}{\overline{u}_\infty} = \left(1 - \sqrt{1 - \frac{C_{\text{T}} \cos \gamma}{8\sigma_y \sigma_z / d^2}}\right) \exp\left(-0.5\left(\frac{y - \delta}{\sigma_y}\right)^2\right)
$$
$$
\exp\left(-0.5\left(\frac{z - z_h}{\sigma_z}\right)^2\right), \tag{2}
$$

where $\Delta \overline{u}/\overline{u}_\infty$ is the velocity deficit in the wake; $C_{\text{T}}$ is the thrust coefficient; $\gamma$ is the yaw angle, which is assumed to be 0 throughout this paper; $y - \delta$ and $z - z_h$ are the distances from the wake center and the point of interest in the cross-stream horizontal and vertical directions, respectively; and $\sigma_y$ and $\sigma_z$ are the standard deviations of the wake deficit, again in the cross-stream horizontal and vertical directions, respectively. These standard deviations are defined in Eqs. (3) and (4).

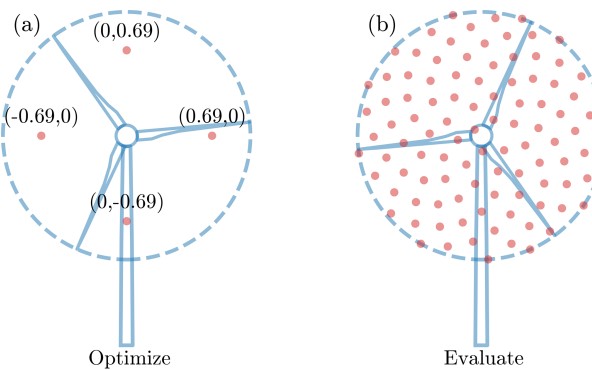

**Figure 5.** The sampling points across the swept rotor area to calculate the effective wind speed at the turbine. Wind speeds are sampled at each point and then averaged. **(a)** The sparse sampling locations used during optimization. The coordinates shown are normalized by the rotor radius. **(b)** The 100 sample points used for final evaluation.

$$\sigma_y = k_y(x - x_0) + \frac{D \cos \gamma}{\sqrt{8}}, \tag{3}$$

$$\sigma_z = k_z(x - x_0) + \frac{D}{\sqrt{8}}, \tag{4}$$

where $D$ is the diameter of the wind turbine creating the wake, $x - x_0$ is the distance downstream from the turbine to the point of interest, and $k_y$ and $k_z$ are unitless and are functions of the free-stream turbulence intensity:

$$k_y, k_z = 0.3837 \, \mathrm{TI} + 0.003678. \tag{5}$$

Because $\gamma = 0$ throughout this paper, $\cos(\gamma) = 1$ meaning that $\sigma_y = \sigma_z$. Wakes were combined with a linear combination method, about which more details can be found in the cited literature (Bastankhah and Porté-Agel, 2016; Thomas and Ning, 2018).

To find the effective wind speed across the entire wind turbine to be used in turbine power calculation, we averaged the velocities sampled at several points across the rotor. During optimization, we sampled at four points over the swept area of the rotor, shown in Fig. 5a. We have found that using just these four sampling locations gives an almost identical effective velocity compared to using more sampling points. For the final evaluation, we sampled the wind speed at 100 points equally spread across the rotor swept area, shown in Fig. 5b.

### 3.4 Wind resource

As the goal of this paper is to demonstrate the performance of our layout parameterization method in wind farm optimization for any scenario, we chose three different wind roses from cities in California, USA: North Island, Ukiah, and

Victorville.[2] During optimization, we divided the wind roses into 24 equal bins for each wind rose, with an associated directionally averaged wind speed, shown in Fig. 6. We have assumed that the wind speed distribution from each wind direction can be approximated with a Weibull distribution defined with the directionally averaged wind speeds (Fig. 7 and Eq. 6). Weibull distributions have been shown to be good representations of real wind speed data (Justus et al., 1978; Rehman et al., 1994; Seguro and Lambert, 2000) [TS2]

$$f(U, \lambda, k) = \frac{k}{U_{\mathrm{mean}}} \left(\frac{U}{U_{\mathrm{mean}}}\right)^{k-1} e^{-(U/U_{\mathrm{mean}})^k}$$

$$\lambda(U_{\mathrm{mean}}, k) = \frac{U_{\mathrm{mean}}}{\Gamma(1 + 1/k)} \tag{6}$$

In Eq. (6), $f$ is the probability of wind for a given wind speed, $U$ is any wind speed (non-negative), $U_{\mathrm{mean}}$ is the directionally averaged wind speed for the direction bin of interest, and $\Gamma$ is the gamma function. The shape parameter, $k$, is assumed to be equal to 2.0 for every wind direction, which is a realistic value for the Weibull distributions that represent real wind speed probability data (Rehman et al., 1994; Seguro and Lambert, 2000). For each wind direction, we have sampled the Weibull distribution at five equally spaced points during optimization. Five wind speed samples and 24 wind direction samples are chosen as the sampling amount required to converge to the true wind farm production for a given wind farm (Stanley and Ning, 2019). Although the wind farms are optimized with the coarser sampling of 24 wind directions and 5 wind speeds, the final wind farm layouts are evaluated with a finer sampling of 360 wind directions and 50 wind speeds, to avoid the possibility of artificially inflated energy production due to coarse wind resource sampling.

## 4 Optimization

In this paper we compare how optimizing with BG wind farm layout parameterization compares to two common currently used parameterization methods. We have optimized wind farms using a simple grid parameterization (referred to as "grid optimization") and BG parameterization ("BG") and by directly optimizing the location of each turbine independently ("direct optimization"). Examples of these layouts, along with the baseline layout that was used to compare results in Sect. 5.1, are shown in Fig. 8.

In each case, the objective function of the optimization was to maximize the annual energy production (AEP) of the wind farm, shown in Eq. (7). [TS3]

$$\mathrm{AEP} = 8760 \sum_{i=1}^{23} \sum_{j=1}^{5} P\left(\phi_i, U(\phi_i)_j\right) f_i f_j \tag{7}$$

---

[2]https://mesonet.agron.iastate.edu/sites/windrose.phtml?station=AAT&network=CA_ASOS (last access: 9 December 2019).

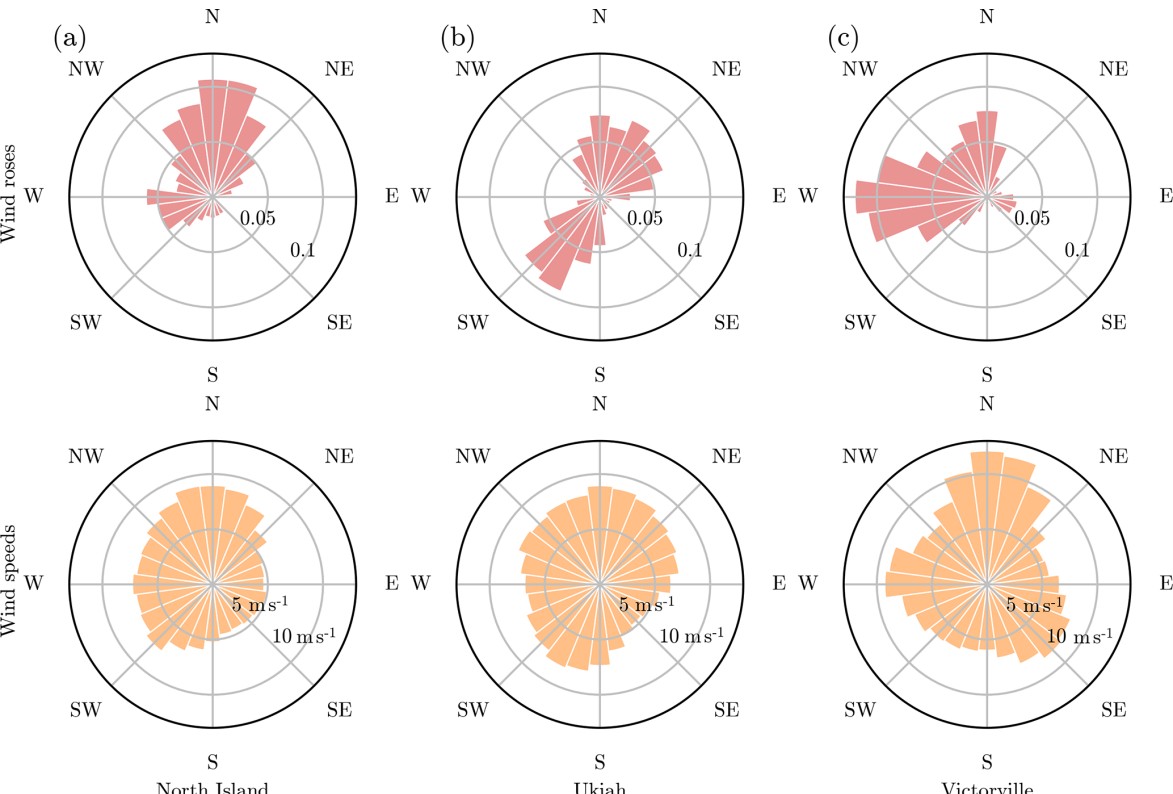

**Figure 6.** The three wind roses and associated average wind speeds used in this study. The wind resources are from **(a)** North Island, California, **(b)** Ukiah, California, and **(c)** Victorville, California.

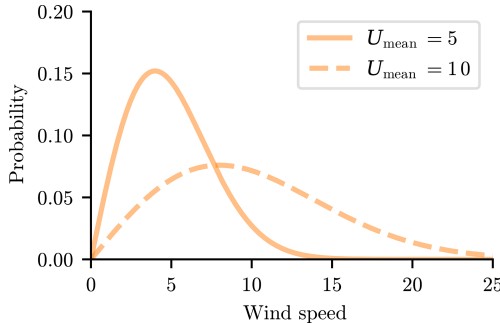

**Figure 7.** Example Weibull distributions for two different average wind speeds. Each wind direction is associated with an average wind speed (shown in Fig. 6), which is used for the value $U_{mean}$.

In this equation, 8760 is the number of hours in a year, $P$ is the wind farm power production, $\phi$ is the wind direction, $V$ is the free-stream wind speed, $f_i$ is the wind direction probability, and $f_j$ is the wind speed probability. The design variables were determined by the optimization method that was used. For the grid optimization, the design variables were the grid spacing in the $x$ and $y$ directions, d$x$ and d$y$, the grid offset $b$, and the grid rotation $\theta$ for a total of four variables. The discrete variables in the grid were determined with the

same method described above to find the discrete variables in the grid portion of the BG parameterization, except d$y = $ d$x$ or d$y = $ 2d$x$ while determining the grid format. We experimented with different values of d$y$ during grid initialization and found that the 1 : 1 or 1 : 2 ratios provided the best results. We ran every grid optimization with each initialization ratio and chose the best results. The design variables for the BG optimization were the same as the grid optimization for the inner grid turbines and an additional variable $s$ defining the start location of the boundary turbines for a total of five design variables. For the direct optimization methods, the design variables were the $x$ and $y$ locations of each turbine in the wind farm for a total of 200 design variables. In each optimization, we applied turbine spacing constraints and boundary constraints. The turbine hub locations were constrained to not be within 2 rotor diameters of any other turbine hub. Additionally, the turbine hubs were constrained to be within the defined wind farm boundary. No bound constraints or additional constraints were used to define where the turbines must lie. A link for the code used in this project is included at the end of this paper. Please refer to the code for specific details about how these constraints were enforced. This optimization is expressed in Eq. (8).

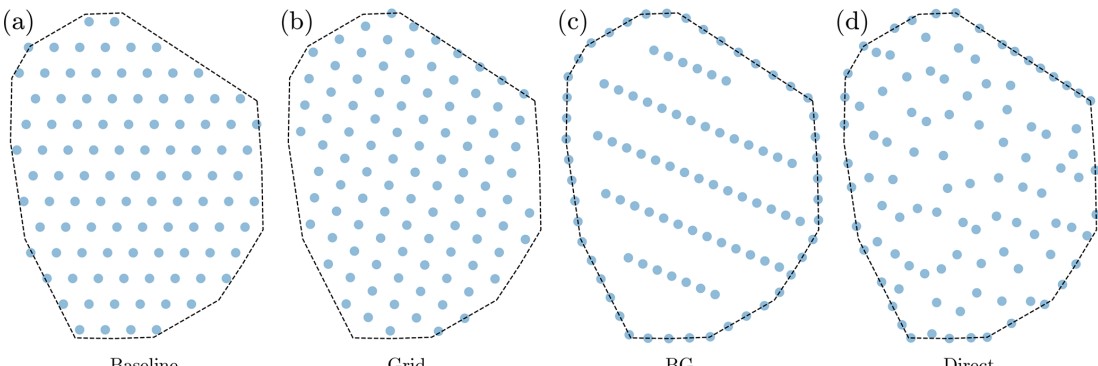

**Figure 8.** Example optimal layouts achieved with each parameterization method. These are 100-turbine layouts, with an average turbine spacing of 4 rotor diameters and the Princess Amalia wind farm boundary. They were optimized with the wind rose from North Island, California. **(a)** The baseline grid to which other methods were compared in Sect. 5.1. **(b)** An example optimized grid layout. **(c)** An example optimized boundary-grid layout. **(d)** An example layout that was optimized directly.

$$
\begin{array}{lll}
\text{maximize} & \text{AEP} & \\
\text{with regard to} & \mathrm{d}x, \mathrm{d}y, b, \theta & \text{(grid)} \\
& \mathrm{d}x, \mathrm{d}y, b, \theta, s & \text{(BG)} \\
& x_i, y_i (i = 1, \ldots, 100) & \text{(direct)} \\
\text{subject to} & \text{boundary constraints} & \\
& \text{spacing constraints} &
\end{array} \qquad (8)
$$

We used the optimizer SNOPT, which is a gradient-based optimizer that uses sequential quadratic programming and is well suited to large-scale nonlinear problems such as the wind farm layout optimization problem (Gill et al., 2005). A challenge of gradient-based optimization is the tendency to converge to local solutions. In order to better search design space, we optimized the problem to convergence 100 times with randomly initialized design variables. The random initialization was performed by fully randomizing the rotation variable $\theta$ and the boundary start location $s$ and defining the discrete and other design variables as defined in Sect. 2.2. The design variables $\mathrm{d}x, \mathrm{d}y$, and $b$ are then randomly perturbed by plus or minus 10 %. This random initialization method allows the number of rows and columns in the inner grid to differ between optimization runs. This was done for each parameterization method, lending confidence that the best solution after optimizing the 100 random starts is near the global optimum. From the random starting points, we were also able to determine the spread of solutions obtained with each layout parameterization.

We used exact-analytic gradients in each optimization. The gradients for each portion of the model were obtained with an automatic differentiation source code transformation tool, Tapenade (Hascoet and Pascual, 2013). To combine the gradients to get the total derivative of the objective with respect to each of the design variables, we used the open-source optimization framework, OpenMDAO, which propagates the partial derivatives of each small section of the model and calculates the gradients of the entire system (Gray et al., 2010).

Using exact, rather than finite-difference, gradients is important in this study because the computational expense required for optimization problems with increasing design variables scales better with exact gradients (see Fig. 2). For the parameterized optimizations, the exact gradients were not as vital in terms of computational expense, but they were very important for the direct optimizations which had 200 design variables. In addition to reducing the function calls required to reach convergence, the exact gradients helped the optimizer converge to a better solution, avoiding many of the numerical difficulties that often plague the optimization process when using finite-difference gradients.

For this paper we have used only a gradient-based optimization method. The purpose of this research is to explore a novel wind turbine layout parameterization and how it compares to other more commonly used layout parameterizations. We do not explore how different optimization methods compare when applied to the wind farm layout problem. As mentioned in the Introduction, the relationship of how optimization method performance scales with increasing numbers of design variables is well documented. Additionally, our past work suggests that the large number of random starts allows for a reasonably thorough search of the design space.

## 5 Results and discussion

In this section we demonstrate how the optimal wind farms using BG parameterization compared to wind farms that have been optimized directly or with a common grid parameterization. We will discuss the best results, the computation expense required to optimize and the multimodality of the design space with each parameterization method.

### 5.1 Best results

Figure 9 shows the best results of the 100 random starts for each parameterization method, compared to a simple base-

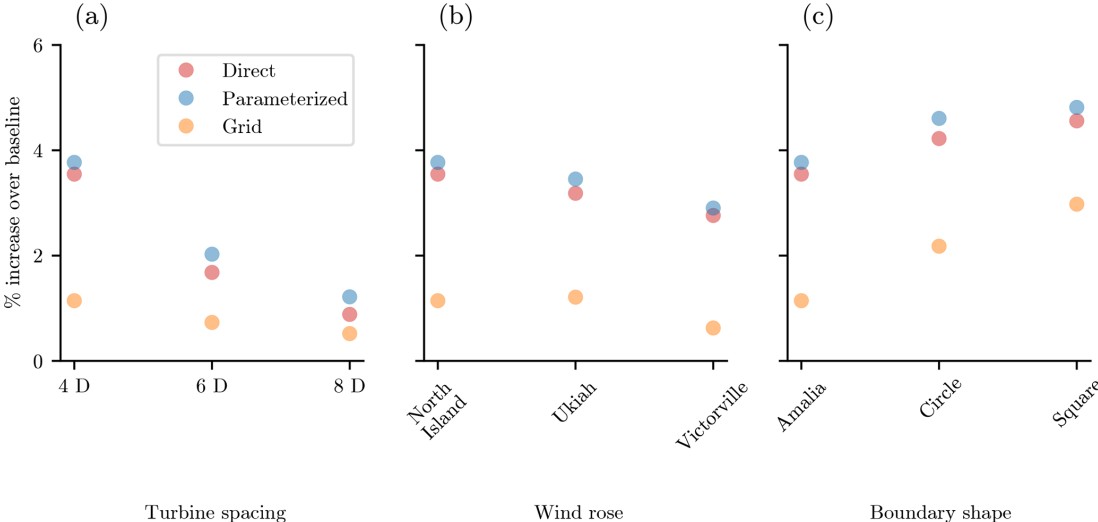

**Figure 9.** The best annual energy production achieved with 100 randomly initialized optimizations. Shown are the best results from the grid turbine parameterization (four design variables), our new boundary-grid parameterization method (five design variables), and by directly optimizing the location of each turbine (200 design variables). Results are shown as a percent increase over a baseline grid layout. **(a)** Varied average turbine spacing in the wind farm. **(b)** Varied wind rose. **(c)** Varied boundary shape.

line grid (Fig. 8a). In Fig. 9, panels (a), (b), and (c) show results for varied turbine spacing, wind roses, and boundary shapes, respectively. For each wind farm BG layout parameterization performs slightly better than the direct layout optimization, although all BG results are within 0.4 % of the corresponding direct results. This does not mean that directly optimizing the layout of each turbine cannot perform as well as the BG parameterization. Clearly, with complete freedom of where to place each wind turbine, the optimizer could find the exact same layout as the BG layout. However, the complete freedom of the direct optimization means that the optimizer is free to explore many suboptimal layouts as well and will often converge in those areas. With BG parameterization, we have forced the turbines to only explore desirable turbine locations. For the scenarios that we explored, 100 BG optimizations produced a better result than 100 direct optimizations.

Figure 9a shows the optimal results for wind farms with varied average turbine spacing, with the North Island wind rose and Princess Amalia wind farm boundary. For the smallest, most tightly packed wind farm, the optimized grid performs better than the baseline but underperforms by about 2.3 % compared to the other parameterization methods. Even at an average turbine spacing of 6 rotor diameters, the direct and parameterized optimizations perform about 1 % better than the grid optimization, which may or may not be significant depending on the uncertainty of the models used. For the largest wind farm, the optimal grid performs within 0.4 % of the other parameterization methods. For large wind farms where the turbines are spaced very far apart, wakes are mostly recovered by the time they reach other turbines in the

wind farm. In these cases, even an optimized grid performs almost as well as the direct or BG optimization.

Figure 9b shows results for optimized wind farms with different wind resources, with an average turbine spacing of 4 rotor diameters and the Princess Amalia wind farm boundary. The wind roses and the associated directionally averaged wind speeds are shown in Fig. 6. As with the varied turbine spacing results, the BG results are slightly better than the direct optimizations and much better than the simple grid. For each wind rose, the grid achieves a slight improvement over the baseline but underperforms by 2 %–2.3 % compared to the direct and BG parameterizations.

Figure 9c shows the results for a varied wind farm boundary. The farms in this subfigure have an average turbine spacing of 4 rotor diameters and the North Island wind rose. Consistent with the previous results, the parameterized optimization performs superbly, always slightly outperforming the direct optimizations. In addition, we can see that the BG and direct optimizations perform better than the simpler grid optimizations, by 1.5 %–2.3 %.

In terms of the best achievable wind farms with each parameterization method, our new BG method performs almost identically to optimizing the location of each wind turbine directly. In all cases that we tested, the BG optimizations were able to find solutions that slightly outperformed the direct optimizations, although they were almost identical. With only five design variables, we can create wind farms that perform the same as or better than farms that have been designed with 200 variables. While the grid parameterization is able to achieve good results for some wind farms, it often performs much worse than our parameterization. One additional vari-

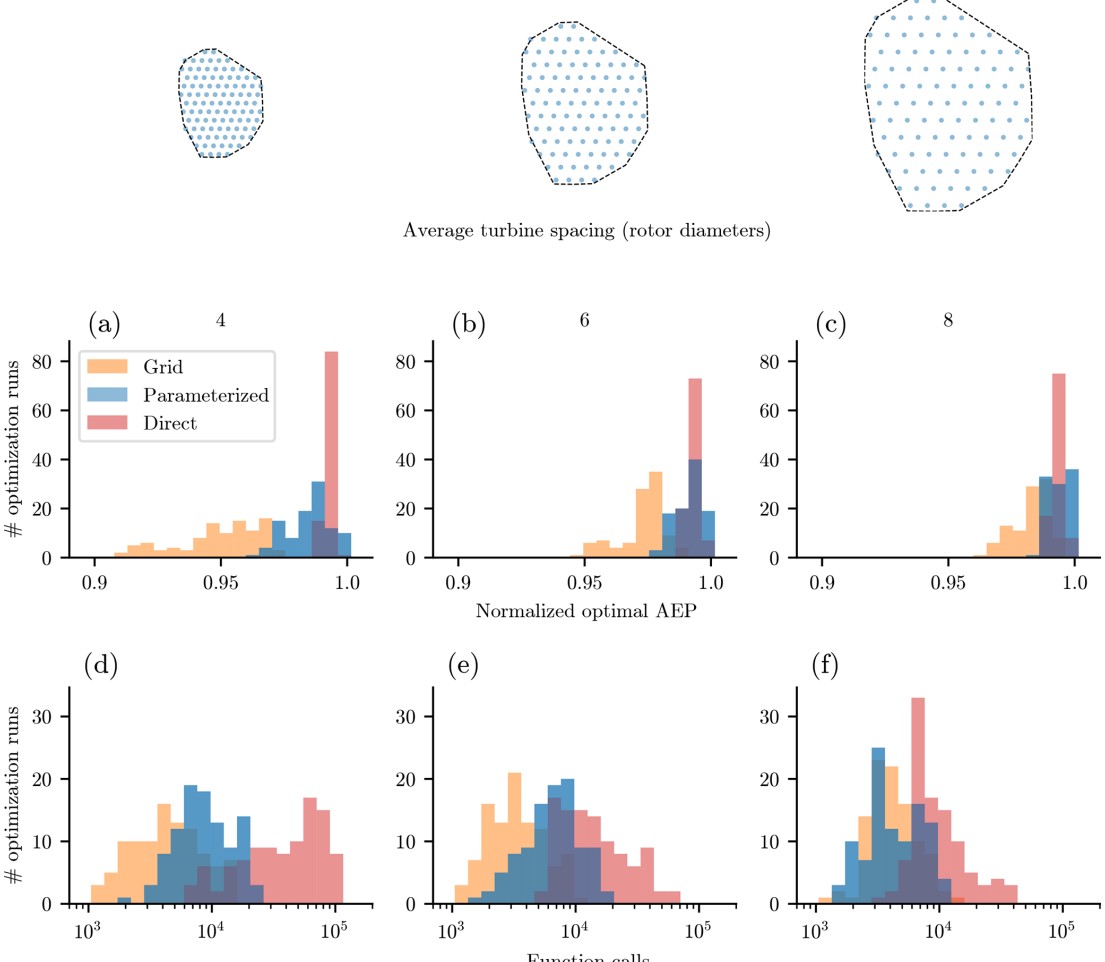

**Figure 10.** Results from 100 randomly initialized optimizations for wind farms with varied average turbine spacing and 100 wind turbines. The farm optimized had the Princess Amalia boundary and the wind rose from North Island, California. Shown are results using the grid turbine parameterization, our new boundary-grid parameterization, and direct optimization. The optimal annual energy production distribution achieved for each of the optimization runs, in wind farms with varied turbine spacing of 4, 6, and 8 rotor diameters for panels **(a)**, **(b)**, and **(c)**, respectively. The number of function calls required to converge for each of the optimization runs, in wind farms with varied turbine spacing of 4, 6, and 8 rotor diameters for panels **(d)**, **(e)**, and **(f)**, respectively.

able is a small price to pay for significant improvement in optimal wind farm design.

## 5.2 Computational expense

The utility of any wind farm layout parameterization is not only measured by the ability to create high energy-producing wind farms, but by the ability to do so quickly and reliably. Figures 10, 11, and 12 are histograms showing optimal results and the computational expense required for each of the 100 optimizations run for each wind farm and parameterization method. In each figure, panels (a)–(c) show the normalized optimal AEP for each of the 100 runs, and panels (d)–(f) show the number of wake model function calls required to converge to a solution. The AEP results have each been normalized by the maximum AEP achieved by the direct op-

timizations for the associated wind farm. Also note that the number of function calls are shown with a log scale.

In general, the grid and the BG optimal AEP results have a similar spread, with the BG results shifted up higher. Compared to the direct optimizations, the grid and BG optimizations have a larger spread in optimal solutions. This is a consequence of the discrete variables that are initialized at the start of each optimization run. The number of rows and columns, as well as their organization in the grid are determined by the randomly initialized rotation design variable, $\theta$. Some of these grid formations are more desirable than others, leading to higher AEP values. This spread in optimal solutions is not a significant issue because the number of functions calls required for the grid and BG optimizations are an order of magnitude lower than that required by the di-

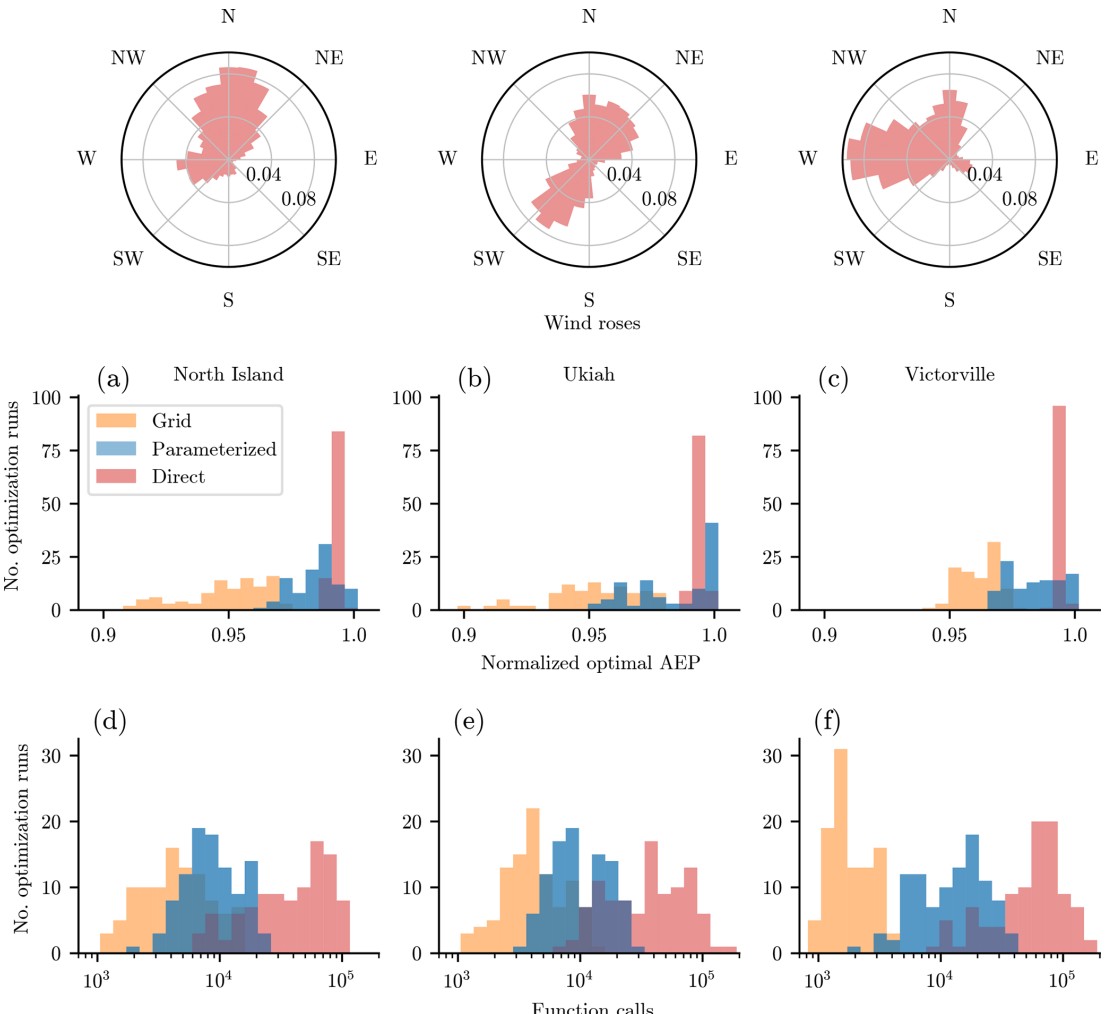

**Figure 11.** Results from 100 randomly initialize optimizations for wind farms with varied wind roses and 100 wind turbines. The farm optimized had the Princess Amalia boundary, and the average turbine spacing was 4 rotor diameters. Shown are results using the grid turbine parameterization, our new boundary-grid parameterization, and direct optimization. The optimal annual energy production distribution achieved for each of the optimization runs, in wind farms with varied wind roses. Wind rose from **(a)** North Island, California, **(b)** Ukiah, California, and **(c)** Victorville, California. The number of function calls required to converge for each of the optimization runs, in wind farms with varied wind roses. Wind rose from **(d)** North Island, California, **(e)** Ukiah, California, and **(f)** Victorville, California.

rect optimization. This allows for many randomly initiated runs in a short amount of time. If it did become an issue, the spread could be reduced by predefining the discrete grid variables or including them as design variables in a gradient-free formulation. By showing the results for three different wind farm sizes, wind roses, and wind farm boundaries, we believe that our parameterization method can produce high AEP and optimize with reduced function calls for many scenarios.

With regards to the function calls required to converge, the grid optimizations required about one-third of the function calls to converge compared to the BG optimizations, while the direct optimizations required about an order of magnitude more. The only exception was the circular wind farm, for which the direct optimizations converged quickly, on the

same order as the BG optimizations. Function calls are an important measure of computational expense, as they are correlated with time and processing power required to optimize. Here it is important to remember that our results were obtained with exact-analytic gradients, meaning that one function call was required to obtain the wind farm AEP as well as the gradients with respect to each of the design variables. The same is true of the constraints: one function call gave both the constraint values and the gradients. Without exact gradients, a finite-difference method would need to be used to calculate the gradients. At every optimization step, finite-difference gradients require one (forward or backward difference) or two (central difference) additional function calls for every design variable to approximate the gradients. Thus,

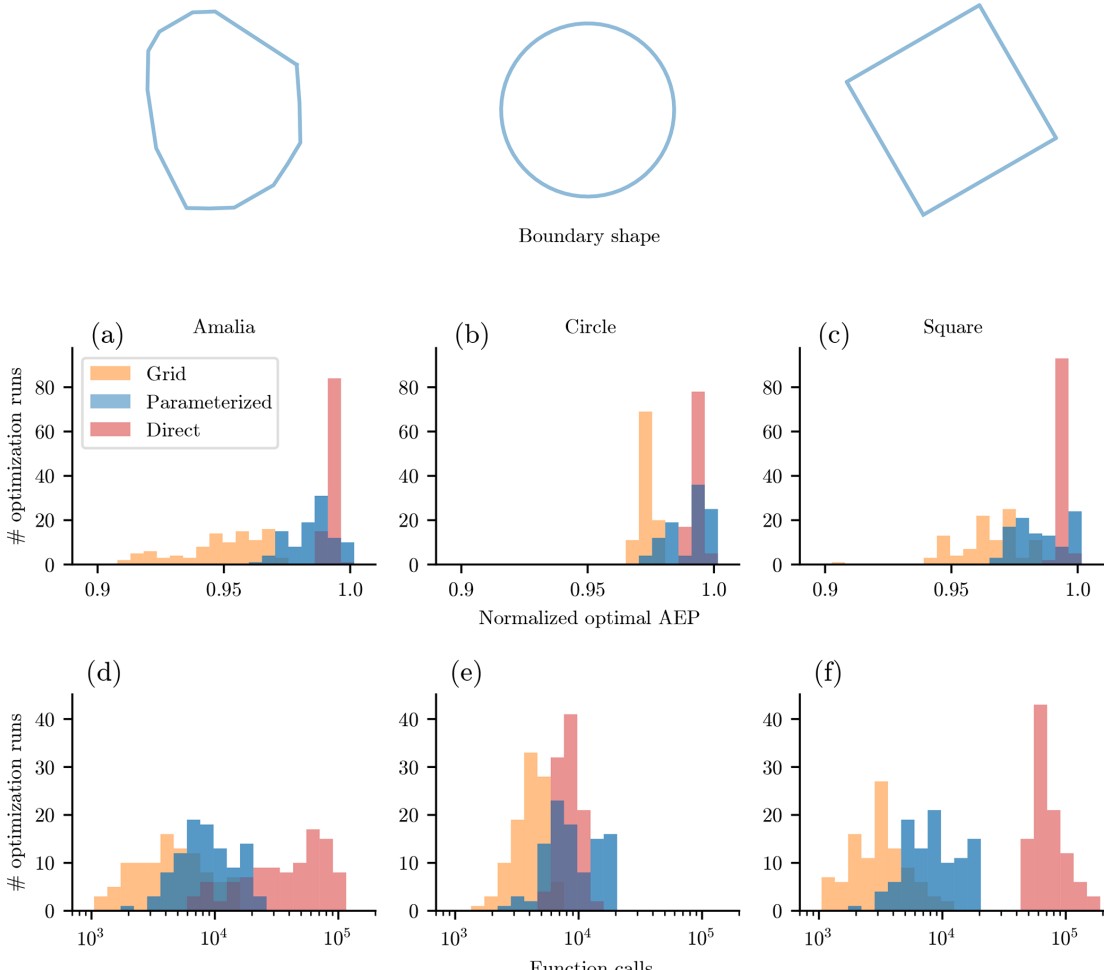

**Figure 12.** Results from 100 randomly initialize optimizations for wind farms with varied wind farm boundaries and 100 wind turbines. The average turbine spacing was 4 rotor diameters, and the wind rose was from North Island, California. Shown are results using the grid turbine parameterization, our new boundary-grid parameterization, and direct optimization. The optimal annual energy production distribution achieved for each of the optimization runs, in wind farms with varied boundary shapes. **(a)** Princess Amalia wind farm boundary. **(b)** Circular wind farm. **(c)** Square wind farm. The number of function calls required to converge for each of the optimization runs, in wind farms with varied boundary shapes. **(d)** Princess Amalia wind farm boundary. **(e)** Circular wind farm. **(f)** Square wind farm.

if forward-difference gradients were used rather than exact ones, the grid optimizations would need about 4 times as many function calls to reach a solution, the BG optimization would need about 5 times as many function calls, and the direct optimization would need 200 times as many function calls to converge. This is the best-case scenario, as optimizations with finite-difference gradients often have trouble converging. Compared to gradient-free optimization, the exact analytic gradients are vital. The direct optimization with a gradient-free technique would be near impossible because of the massive required computational expense (Ning and Petch, 2016; Thomas and Ning, 2018).

## 5.3 Multimodality

One of the major difficulties of the wind farm layout optimization problem is the extreme multimodality of the design space (Fig. 1). There can be thousands or even millions of local solutions, often varying drastically in their quality. Figure 13 shows one-dimensional sweeps across the design variables for each of the three different parameterization methods discussed in this paper. Because of the number of variables in this problem, it is difficult to fully represent the full design space graphically; however, this figure is a good indicator of the multimodality of the different design spaces. Figure 13a, b, and c show the multimodality of the grid, BG, and direct layout parameterizations, respectively.

Parameterizing the design space with a grid and with the BG method (Fig. 13a and b) does not completely remove

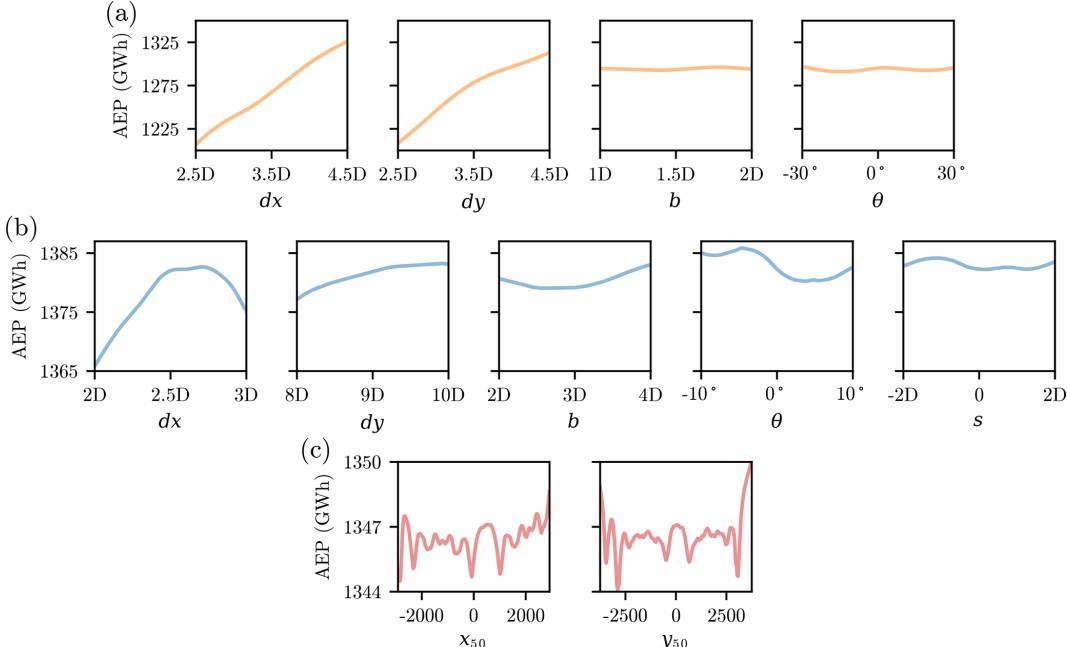

**Figure 13.** One-dimensional sweeps across the design space of each parameterization method discussed in this paper. These figures show the multimodality of each of the design spaces. **(a)** The simple grid parameterization. **(b)** Our newly presented boundary-grid parameterization. **(c)** Moving the location of one wind turbine across the wind farm in $x$ and $y$ (refer to Fig. 1). With the direct turbine layout definition there are actually 200 variables. This figure shows the multimodality in just two of these variables, where the whole design space is much more complex.

the multimodality of the wind farm layout problem. However, it does result in a smoother response and fewer local minima compared to the design space when each of the turbines are optimized directly. These function spaces can be explored easily with a few random starting locations or with a gradient-free optimization method. The design space when varying the location of individual turbines (Figs. 1 and 13c) is much noisier, filled with comparatively larger peaks and valleys in the design space. These figures only show the design space with respect to the location of one turbine, which is defined with two variables. The full space consists of the location of all 100 turbines, or 200 variables, for which the multimodality and overall noisiness of the design space is exacerbated. Figure 13a and b do not show the function space with respect to the discrete grid variables. Even so, considering each combination of the feasible grid variables is more desirable than the difficulty involved with the 200-D function space of the direct layout definition.

Notice that the ranges of the design variable sweeps is different for the BG and grid parameterizations compared to the direct sweep. This is because the simpler parameterizations are more limited in the feasible design values. The range through which the design variables can sweep is relatively limited, without violating the minimum spacing or the boundary constraints.

## 6   Additional details on BG parameterization

BG parameterization requires few variables, produces wind farm layouts that perform similarly to ones that have been optimized directly with much lower computational expense, and reduces the multimodality of the design space. In addition, there are some innate design characteristics that are useful in wind farm design. First, the layouts produced are regular, aesthetically pleasing patterns. To the untrained eye, BG parameterization looks well designed compared to the seemingly random layouts that are often produced when every turbine location is optimized individually. This can play an important role in the public perception of large-scale wind energy. Second, BG parameterization has clear roads or shipping lanes naturally built into the design. Roads and shipping lanes are requirements in wind farm design that are often neglected in research studies.

Often, there are prohibited areas within a wind farm. This could be for many reasons, such as natural geography, roads or shipping lanes, or a variety of other reasons. Although beyond the scope of this paper and not addressed in the results shown in Sect. 5, we have a few ideas on how this would be handled with BG parameterization. Many prohibited zones, such as shipping lanes, roads, or cable lines, are easily managed with a grid turbine layout, as these could easily be designed to follow the existing grid layout. Other prohibited zones could be handled by the BG parameterization, with no

adjustments. This would be for cases where the prohibited zones are relatively small. For other cases, where the prohibited zones are larger and more restrictive, slight modifications would need to be made to the parameterization. The discrete variable of the inner grid would be initially defined such that the turbine location constraints are met. This would likely include some of the rows that are not continuous, but have some gaps to accommodate the constraints. Likewise, the boundary turbines would be defined slightly differently, in that there would be some gaps to accommodate layout constraints.

## 7 Conclusions

In this paper, we have presented the new boundary-grid wind farm layout parameterization method. This method uses only five design variables, regardless of the number of wind turbines but is capable of producing turbine layouts that perform just as well as or better than layouts where the location of each wind turbine has been optimized directly. We optimized the layout of seven different wind farms with three different parameterization methods: a simple grid, directly optimizing the location of each turbine, and our new boundary-grid parameterization. For each wind farm and parameterization method, we ran 100 optimizations with randomly initialized design variables. In every case, the best layout achieved with the BG parameterization perform slightly better than the best layout achieved with the direct optimizations.

In addition to being able to match the optimal energy production of wind farms that were directly optimized, BG parameterization requires an order of magnitude fewer function calls to reach a solution. This is with exact-analytic gradients, which means if finite-difference gradients or a gradient-free optimization method were used instead, our parameterization method would require at least 2 to 3 orders of magnitude fewer function calls to optimize. BG parameterization also reduces the multimodality of the design space, simplifying the optimization process and making it easier to find a good solution.

The BG layout definition places a portion of the wind turbines around the boundary, spaced equally traversing the wind farm perimeter. The rest of the turbines are placed in a grid inside the farm boundaries. The wind farm layouts created have a regular, aesthetically pleasing pattern, naturally defined roads and shipping lanes, and an easily defined cabling pattern. BG parameterizations solve many of the problems that typically accompany wind farm layout optimization. It is a simple, easily implemented technique that can immediately be applied by researchers and wind farm developers, playing an important role in the continued growth of wind energy.

**Code and data availability.** The code written for this paper is included at https://doi.org/10.5281/zenodo.3523383 (Stanley, 2019).

All dependencies, with the exception of the optimizer SNOPT are open source.

**Author contributions.** APJS led this research, including designing and testing potential parameterization methods, running the optimizations, and writing the paper. AN helped develop ideas and methodology, provided feedback throughout the entire process, and provided editing for the paper.

**Competing interests.** The authors declare that they have no conflict of interest.

**Review statement.** This paper was edited by Rebecca Barthelmie and reviewed by Sebastian Sanchez Perez-Moreno and Ju Feng.

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

Please note the remarks at the end of the manuscript.

## Remarks from the typesetter

TS1    According to our standards, changes like this must first be approved by the editor, as data have already been reviewed, discussed and approved. Please provide a detailed explanation for those changes that can be forwarded to the editor. Please note that this entire process will be available online after publication. Upon approval, we will make the appropriate changes. Thank you for your understanding.

TS2    Please see previous comment regarding editor's approval.

TS3    Please see previous comment regarding editor's approval.

TS4    Please provide date of last access.

TS5    Is there are link to access this reference?

TS6    Is the first author Singg or Zingg?

TS7    Is this the same author as in the previous references (Stanley, A. P. J.)?

TS8    Please provide date of last access.

TS9    Please provide date of last access.