# Peer review of "Massive Simplification of the Wind Farm Layout Optimization Problem"

_Wind Energy Science, 2019_

## Referee Comment (RC1) · Sebastian Sanchez Perez-Moreno (Referee) · 25 Aug 2019

The reviewer very much appreciates the effort in presenting results in a clear, concise, and visually appealing way, as well as the availability of the computational codes. These two efforts contribute enormously to the understanding, and reproducibility of the results, which should set a precedent to all authors of this journal.

Regarding content, in spite of the massive oversimplification of the layout optimisation that could quickly lead to infeasible designs due to the high number of constraints industry faces in practice, it is very valuable to see that AEP-wise the direct and parameterised approaches are not that different. It is furthermore acknowledged that this academic effort to benchmark three design procedures so robustly with three different

energy densities, shapes and windroses, provides high value and further evidence of AEP behaviour for this tremendously complicated optimisation problem.

Nevertheless, there are a few points for discussion, that should further improve the understanding of the proposed procedure and make the approach more transparent as well.

The optimal distribution of the turbines on the site boundary will depend on the windrose and shape of the site. Have you tested a more sophisticted algorithm to fill the perimeter with variable spacing according to wind direction and direction of the sites edges? Is a higher AEP expected than if placing them using a uniform spacing?

The authors suggest placing 45% of turbines on the boundary, when feasible. This sounds too case-specific. While I understand that the gradient-based optimisation algorithm requires a smooth function, and that letting the optimiser vary the spacing of the turbines on the perimeter and thus moving turbines inside the site would lead to "jumps" in the AEP response surface, I believe that fixing the number of turbines arbitrarily does not help the design space either. Would you suggest to re-run your method with different spacings/number of turbines on the perimeter? Is this done at all in the 100 randomly initialised runs of section 5? Why not make the spacing the design variable, and let the number of turbines on the boundary be variable. AEP surface would be too discontinuous?

A constant CT is assumed by the wake modelling, is there a noticeable difference in AEP compared to using a Ct curve?

During the initialisation procedure suggested, dy is 4 times dx, is there empirical evidence for it? However, if I understand correctly, dy is varied later to fit the desired number of turbines inside the site area, is this initial ratio not lost then?

Also, the b variable is initialised to offset rows by 20 deg, is there a reason for this seemingly arbitrary value? Why not stagger rows by one half dx?
The initialisation procedure is meant to fix the number of rows and columns across the optimisation. The last paragraph of section 2.2 implies that the optimisation does not allow turbines to "jump" between rows, or to trade columns for rows. Is this what varies between the 100 runs of section 5?

How are the authors checking which turbines are inside the area? Can you share what algorithm you are using for that matter?

How are the authors defining the inner area in which the grid turbines must lie? Is there a uniform buffer spacing from the perimeter enforced?

How do the authors foresee they will deal with prohibited zones inside the area?

How are turbines placed along the perimeter? Is there consideration that two turbines near a corner could be closer than the minimum desired spacing?

Yes! The following text was added to section 2.2 to clarify this:

"When defining the number of turbines to be placed along the perimeter, the user must consider the most extreme boundary angles, such that minimum turbine spacing is preserved even at boundary corners."

What can be said of the results in Fig. 8 with respect to farm energy density? And in general, do the similar AEP results hold for all area densities?

The results in Figure 8 are intended to show that yes, we expect similar AEP results between the direct and parameterized optimizations regardless of the farm energy density. The following was added to the paper in Section 5.2:

"By showing the results for 3 different wind farm sizes, wind roses, and wind farm boundaries, we believe that our parameterization method can produce high AEP and optimize with reduced function calls for any scenario."

How would the authors deal in cases where all the internal and perimeter turbines have to align to an underlying base grid, for shipping and rescue operations?

What are the differences exactly between the 100 runs of the parameterised optimisation, the initial values of all variables? Different number of rows and columns? Or just the orientation angle theta?

Finally, is there future work aligned with this one? Are more/different variables interesting to look at for the design of wind farm layouts?

Technical correction: I suggest changing "verses" for "versus" in more than one place (e.g. line 25, fig 2).
* * *

---

## Referee Comment (RC2) · Ju Feng (Referee) · 28 Aug 2019

This paper proposed an interesting parameterization method for wind farm layout optimization, that has the potential of largely reducing the number of design variables. In general, the paper is well written, the new method is useful and results seems promising.

However, there are some major concerns the reviewer has on the current paper that he recommend this paper for a major revision. The major concerns are as follows:

1. Missing details in the proposed boundary-grid parameterization

As the central contribution of this study, the boundary-grid parameterization is not presented in a complete and clear manner. After reading Section 2, the reviewer can't

figure out how exactly the 5 design variable can determine one and only one layout inside the specified boundary with a given number of turbines. For example, if dx and dy is too big, the number of turbines you can put in the inner grid will be very few, then there might be too many turbines placed on the perimeter, that violate the minimal spacing constraints. Also the same set of dx, dy, theta and b can define a set of grid points that actually shift in the boundary, which will correspond to different layout. So the reviewer would argue that dx, dy, theta and b alone can't have a one-to-one map to a exact location of grid point.

The selection of discrete values also seems a little bit arbitrary. It is stated in lines 87-88, the discrete values remain fixed, but then again, you have the situation that there are too many grid points inside the boundary (when dx and dy are small), if you have to put 45% turbine around the boundary, you will have to remove some grid points, then which ones to remove according to what rule?

2. Some shortages in wind farm modelling.

First, in lines 117, it says "the turbulence intensity is equal to 0.0325", but shouldn't turbulence intensity change upon the wind speed?

Second, according to Eqs.(3-4), you use the wake deficit at the rotor center to represent the average wake deficit on the whole rotor, since there is no integration over the rotor area in Eq. (4). This is problematic, as the profile of wake deficit is a Gaussian shape, and the one point deficit in the rotor center could be overestimating the mean deficit, if the two turbines are perfectly aligned.

Third, there are only 5 wind speeds, and 23 wind direction sectors used in the wind resource modelling, according to Eq. (7). It has been shown in some studies that you need finer discretization, for example in (Feng and Shen 2015) in your references. This kind of coarse discretization could give you artificially optimistic AEP gains. You may also check the follow paper for recommended discretization:

[Figure]

Feng, Ju, and Wen Shen. "Modelling wind for wind farm layout optimization using joint distribution of wind speed and wind direction." Energies 8, no. 4 (2015): 3075-3092.

3. The missing of comparison to gradient-free optimization technique.

I understand the focus of this study is on the proposed parameterization. But without direct comparison of the gradient based optimizer to some gradient free ones, e.g., GA or RS, it looks unfounded and somehow biased for a lot of claims that says the gradient free method will be infeasible, or perform worse. Also do you have bounds on the design variables? How are the constraints handled in the optimization process? Penalty function?

4. The calim on the infeasibility of gradient-free technique for large wind farm is unfounded.

AS stated in lines 9-10, Our presented method unlocks the ability to optimize and study large wind 10 farms, something that has been mostly infeasible in the past.". But I found this unfounded, you can check the following paper:

Wagner, Markus, Kalyan Veeramachaneni, Frank Neumann, and Una-May O'Reilly. "Optimizing the layout of 1000 wind turbines." European Wind Energy Association Annual Event 205209 (2011).

Also engineering wake models are very fast to run, it shouldn't become too heavy or even infeasible for a gradient-free optimizer applied to a wind farm with 100 turbines, even if needs 10000 evaluations.

5. Some very relevant references are missing.

Especially studies on grid-like layout optimization. The parameterization for the inner grid has been proposed in a similar way in some studies already. You may find the following two of interest:

González, Javier Serrano, Ángel Luis Trigo García, Manuel Burgos Payán, Jesús

Riquelme Santos, and Ángel Gaspar González Rodríguez. "Optimal wind-turbine micro-siting of offshore wind farms: A grid-like layout approach." Applied energy 200 (2017): 28-38.

Neubert, A., A. Shah, and W. Schlez. "Maximum yield from symmetrical wind farm layouts." In Proceedings of DEWEK. 2010.

Some minor issues:

1. It is stated in lines 20-25 that "Although these methods can be highly effective for small numbers of design variables, the computational expense required to converge scales poorly, approximately quadratically, with increasing numbers of variables. Because of this poor computational scaling, many companies and researchers have been limited in the size of wind farms they can optimize, as the number of variables typically increases with the number of turbines." But I doubt that's the case, since there are already large wind farms be designed and built in the world. Also optimization studies have been conducted for large wind farms, such as Horns Rev 1 with 80 turbines, as in one of your references (Feng and Shen, 2015).

2. Lines 31-32 "Power losses of 10–20% are typical from turbine interactions within a wind farm (Barthelmie et al., 2007, 2009; Briggs, 2013), and can be as high as 30–40% for farms with closely spaced wind turbines (Stanley et al., 2019)." This is somehow misleading, power losses of 30-40% are the worst wake case, which doesn't happen that frequent in reality. So the actually AEP loss due to wake effects should be usually lower than 10-20%.

3. Rosenbrock function is used to demonstrate the convergence of gradient based optimizer scales better than gradient-free methods. First, you need to show what is Rosenbrock function, or at least provide a reference. Second, this function is a function that we actually know where are the optimums, thus, we can easily see when it has converged to a local minimum. But in real life applications, we often can't analytically prove that we have reached a local minimum, such as in layout optimization. Third, for

such problem, converge faster (typically for gradient based methods) is just one aspect, the other aspect is the quality of the optimized results, i.e., whether the solution is close to the global optimum. Usually it is know that gradient free methods converges slower but has a higher probability to reach the global optimum, while gradient based methods converge faster but are also easier to be trapped in a local minimum.

4. Eq. (6), U_mean should be scale factor of the Weibull distribution. Note that the scale factor is not the same thing as the mean wind speed, instead the mean wind speed should be a function of scale factor and shape factor.

This will be corrected in the revised manuscript. Final results will include this correction.

5. Line 275-276 states that "BG parameterization, cabling requirements can be clearly minimized by running cables across each of the rows, and around the boundary without the need for complex cabling algorithms". This is not true, as you still need to decide the location of sub-station, the exact topology of the cables and select cable types for different connections, thus, not necessarily easier than any random layout.

---

## Author Comment (AC1) · 2 Oct 2019

**Response to Reviewer 1**

Andrew PJ Stanley and Andrew Ning

October 2, 2019

Thank you for your thorough review of the manuscript and for your comments! We will address each of your comments and questions individually.

Question/Comments are in black.

The corresponding responses are immediately below in blue.

The reviewer very much appreciates the effort in presenting results in a clear, concise, and visually appealing way, as well as the availability of the computational codes. These two efforts contribute enormously to the understanding, and reproducibility of the results, which should set a precedent to all authors of this journal.

Regarding content, in spite of the massive oversimplification of the layout optimisation that could quickly lead to infeasible designs due to the high number of constraints industry faces in practice, it is very valuable to see that AEP-wise the direct and parameterised approaches are not that different. It is furthermore acknowledged that this academic effort to benchmark three design procedures so robustly with three different energy densities, shapes and windroses, provides high value and further evidence of AEP behaviour for this tremendously complicated optimisation problem.
Nevertheless, there are a few points for discussion, that should further improve the understanding of the proposed procedure and make the approach more transparent as well.

The optimal distribution of the turbines on the site boundary will depend on the windrose and shape of the site. Have you tested a more sophisticted algorithm to fill the perimeter with variable spacing according to wind direction and direction of the sites edges? Is a higher AEP expected than if placing them using a uniform spacing?

Yes, we have tested the parameterization where the boundary turbines are spaced equally perpendicular to the dominant wind direction. The idea was that this would avoid layouts where turbines are very close together parallel to the dominant wind direction. However, with the cases we tested we found that equally spacing the boundary turbines around the perimeter performed better. The following text will be added to Section 2.1 to explain this:

"During development of our parameterization method, we tested various strategies of spacing the turbines around the boundary. However, we found that equally spacing the turbines around the perimeter consistently provided the best results."

The authors suggest placing 45% of turbines on the boundary, when feasible. This sounds too case-specific. While I understand that the gradient-based optimisation algorithm requires a smooth function, and that letting the optimiser vary the spacing of the turbines on the perimeter and thus moving turbines inside the site would lead to "jumps" in the AEP response surface, I believe that fixing the number of turbines arbitrarily does not help the design space either. Would you suggest to re-run your method with different spacings/number of turbines on the perimeter? Is this done at all in the 100 randomly initialised runs of section 5?

This is an excellent point. Yes, 45% is a specific number which may be slightly sub-optimal for some specific cases. At length, we looked into the performance of BG parameterization for different numbers of turbines on the perimeter for different wind resources, wind farm boundary shapes and sizes. Our original goal was to find a relationship between some non-dimensional wind farm metric and the best ratio of turbines to place on the boundary. However, in every case we considered, placing 45% of the turbines performed the best or very close to the best compared to other amounts. This consistent good performance, along with the simplicity of having a this number as a constant led us to recommend the number of turbines of the boundary as a constant 45%.

Given sufficient computational resources, yes we would suggest this. However, if resources or time is limited, we would suggest using 45%. The following paragraph was added to section 2.2 to explain this:

"The process outlined to select the discrete variables used in the parameterization is recommended as a starting point, and when computational resources or time is limited. We tested many different methods of how to determine the discrete values, but found that the method shown above consistently produced wind farm layouts with high energy production. With sufficient resources, some scenarios may benefit from optimizing with a different ratio of boundary turbines, or different initializations of the boundary grid. However, the results discussed in this paper were produced with the method given in this section."

Why not make the spacing the design variable, and let the number of turbines on the boundary be variable. AEP surface would be too discontinuous?

Exactly. Discrete variables are not favorable for a gradient-based optimization. If desired, a user could certainly include the number of perimeter turbines as a design variable with a gradient-free approach. The following text was added to section 2.2

discussing this:

"Because these variables are discrete, they cannot be included as design variables when using a gradient-based optimization method, because the function space would be discontinuous. But, a gradient-free optimization may benefit from including some of these discrete variables as design variables in the optimizations."

A constant CT is assumed by the wake modelling, is there a noticeable difference in AEP compared to using a Ct curve?

The AEP with a constant CT is lower than that with a CT curve. A constant CT does not reduce the thrust after rated power is reached, making the predicted wakes stronger than reality. Although not of vital importance to the purpose of our paper, we are already rerunning the wind farm optimizations to make a correction in the mean wind speed, so the results of our revised submission will include a CT curve.

During the initialisation procedure suggested, dy is 4 times dx, is there empirical evidence for it?

The authors tested several different initialization methods for dy, and this method gave the most consistent good results. For some specific cases, a different initialization method may be desirable. However, for the cases we tested, this provided the best results overall. We added text to section 2.2 discussing this (shown in a response above).

However, if I understand correctly, dy is varied later to fit the desired number of turbines inside the site area, is this initial ratio not lost then?

Correct. This ratio only applies to the initialization of the discrete design variables, which are adjusted during optimization.

Also, the b variable is initialised to offset rows by 20 deg, is there a reason for this seemingly arbitrary value? Why not stagger rows by one half dx?

The authors agree, there is some arbitrariness to the initialization of $b$. We tested several different combinations of discrete variable selection, and included the rules that provided the most consistent and best results for us. Although for specific cases there may be a better method, in general the rules we provide worked well. Again, the paragraph we added for the revised paper in section 2.2 (shown in a response above) discusses this.

The initialisation procedure is meant to fix the number of rows and columns across the optimisation. The last paragraph of section 2.2 implies that the optimisation does not allow turbines to "jump" between rows, or to trade columns for rows. Is this what varies between the 100 runs of section 5?

That is correct. The following text will be added to Section 4 to clarify this idea:

"The random initialization was performed by fully randomizing the rotation variable $\theta$ and the boundary start location $s$, and defining the discrete and other design variables as defined in Sec. 2.2. The design variables $dx, dy$, and $b$ are then randomly perturbed by plus or minus 10%. This random initialization method allows the number of rows and columns in the inner grid to differ between optimization runs."

How are the authors checking which turbines are inside the area? Can you share what algorithm you are using for that matter?
Certainly, we'll give a quick summary here and point to the code where the boundary calculations are made.

The wind farm boundary is defined with a set of sequential points, we assume straight lines between each of the points. Also note that the boundaries that we consider in this work are all convex. For a single turbine to one of the boundary lines:

1. Calculate the unit normal to the boundary line.

2. Calculate the vector defining the perpendicular distance between the turbine and the boundary line.

3. The constraint is then calculated as the dot product of these two vectors.

This is repeated for every turbine, to every boundary line. For a concave boundary, a slightly more complicated algorithm would be necessary, but this suffices for the current work. The boundary constraint code we used can be found here:

10.5281/zenodo.3261037

byuflowlab/stanley2019-variable-reduction/code/position_constraints.py

The function name is `calculate_distance`

We will add a note in the text in Section 4 that a link to the project code is included at the bottom of the paper.

"A link for the code used in this project is included at the end of this paper. Please refer to the code for specific details about how these constraints were enforced."

How are the authors defining the inner area in which the grid turbines must lie? Is there a uniform buffer spacing from the perimeter enforced?

There only thing defining where the inner turbines lie are the boundary and spacing constraints discussed in Section 4. There is no uniform buffer spacing. The following text was added to the revised paper in Section 4:

"No bound constraints, or additional constraints were used to define where the turbines must lie."

How do the authors foresee they will deal with prohibited zones inside the area?

This issue is beyond the scope of the presented research, however we have a few ideas on how this could be addressed. The following paragraph was added to Section 6 discussing this:

"Often, the there are prohibited areas within a wind farm. This could be for many reasons, such as natural geography, roads or shipping lanes, or a variety of other reasons. Although beyond the scope of this paper, and not addressed in the results shown in Sect. 5, we have a few ideas on how this would be handled with BG parameterization. Many prohibited zones, such as shipping lanes, roads, or cable lines, are easily managed with a grid turbine layout, as these could easily be designed to follow the existing grid layout. Other prohibited zones could be handled by the BG parameterization, with no adjustments. This would be for cases where the prohibited zones are relatively small. For other cases, where the prohibited zones are larger and more restrictive, slight modifications would need to be made to the parameterization. The discrete variable of the inner grid would be initially defined such that the turbine location constraints are met. This would likely include some of the rows are not continuous, but have some gaps to accommodate the constraints. Likewise, the boundary turbines would be defined slightly differently, in that there would be some gaps to accommodate layout constraints."

How are turbines placed along the perimeter?

There is no "right answer" as to how to accomplish this, but we can briefly summarize how we accomplished this, then point to the code where we calculate the boundary turbine locations.

Preprocessing:

1. Calculate the perimeter of the wind farm boundary.

2. Divide the perimeter by number of turbines that are desired to have on the boundary, in this paper that was 45% of the total number of turbines. This gives the distance of the perimeter traversed between wind turbines.

3. If this spacing is greater than the minimum desired spacing times $\sqrt{2}$, the preprocessing is finished. If not, reduce the number of turbines until the perimeter traversed between wind turbines is greater than the minimum desired spacing times $\sqrt{2}$. The distance traversed around the boundary is simply the perimeter divided by the number of turbines placed on the boundary. The $\sqrt{2}$ is included to ensure that, except in extreme cases, the minimum turbine spacing is achieved for a convex wind farm boundary (i.e., the most extreme boundary angle would be 90 degrees).

Once the number of turbines and their spacings around the perimeter were determined, the location of each turbine around the perimeter was defined with a single variable, $s$.

1. First, an origin was defined. In our case, this was defined as the first point used to define the wind farm boundary.

2. Second, an "anchor turbine" was placed a distance $s$ along the perimeter from the origin.

3. The remaining turbines were then placed such that all perimeter turbines are spaced equally traversing the wind farm perimeter.

The code is found here:

10.5281/zenodo.3261037

byuflowlab/stanley2019-variable-reduction/code/var_reduction_exact.py

The function name is `makeBoundary`

Is there consideration that two turbines near a corner could be closer than the minimum desired spacing?

Yes! The following text was added to section 2.2 to clarify this:

"When defining the number of turbines to be placed along the perimeter, the user must consider the most extreme boundary angles, such that minimum turbine spacing is preserved even at boundary corners."

What can be said of the results in Fig. 8 with respect to farm energy density? And in general, do the similar AEP results hold for all area densities?

The results in Figure 8 are intended to show that yes, we expect similar AEP results between the direct and parameterized optimizations regardless of the farm energy density. The following was added to the paper in Section 5.2:

"By showing the results for 3 different wind farm sizes, wind roses, and wind farm boundaries, we believe that our parameterization method can produce high AEP and optimize with reduced function calls for any scenario."

How would the authors deal in cases where all the internal and perimeter turbines have to align to an underlying base grid, for shipping and rescue operations?

Refer to our above discussion of prohibited zones within the parameterization.

What are the differences exactly between the 100 runs of the parameterised optimisation, the initial values of all variables? Different number of rows and columns? Or just the orientation angle theta?

The initialization of all of the design variables is randomized. The previously mentioned text we added to section 4 should clarify this:

"The random initialization was performed by fully randomizing the rotation variable $\theta$ and the boundary start location $s$, and defining the discrete and other design variables as defined in Sec. 2.2. The design variables $dx, dy$, and $b$ are then randomly perturbed by plus or minus 10%. This random initialization method allows the number of rows and columns in the inner grid to differ between optimization runs."

Finally, is there future work aligned with this one? Are more/different variables interesting to look at for the design of wind farm layouts?

We do plan to implement the BG parameterization in future layout optimization studies, and perhaps make modifications based on the necessary constraints and design space. However, as for further development of the parameterization, there is no planned work directly associated with this one at the moment.

Technical correction: I suggest changing "verses" for "versus" in more than one place (e.g. line 25, fig 2).

This was corrected.

---

## Author Comment (AC2) · 2 Oct 2019

**Response to Reviewer 2**

Andrew PJ Stanley and Andrew Ning

October 2, 2019

Thank you for your thorough review of the manuscript and for your comments! We will address each of your comments and questions individually.

Question/Comments are in black.

The corresponding responses are immediately below in blue.

This paper proposed an interesting parameterization method for wind farm layout optimization, that has the potential of largely reducing the number of design variables. In general, the paper is well written, the new method is useful and results seems promising.

However, there are some major concerns the reviewer has on the current paper that he recommend this paper for a major revision. The major concerns are as follows:

1. Missing details in the proposed boundary-grid parameterization

As the central contribution of this study, the boundary-grid parameterization is not presented in a complete and clear manner. After reading Section 2, the reviewer can't

figure out how exactly the 5 design variable can determine one and only one layout inside the specified boundary with a given number of turbines.

For example, if dx and dy is too big, the number of turbines you can put in the inner grid will be very few, then there might be too many turbines placed on the perimeter, that violate the minimal spacing constraints.

Thank you for bringing this up, our explanation in section 2.2 may have been lacking. We have added the following text to section 2.2 in order to clarify this:

"Note that the number of boundary turbines is determined before the number of turbines in the inner grid, to ensure that sufficient spacing in maintained between the boundary turbines."

Also the same set of dx, dy, theta and b can define a set of grid points that actually shift in the boundary, which will correspond to different layout. So the reviewer would argue that dx, dy, theta and b alone can't have a one-to-one map to a exact location of grid point.

The following was added to section 2.1 to clarify the parameterization:

"The inner grid is centered around the wind farm center, ensuring a one-to-one mapping from the design variables to the possible wind farm layouts."

The selection of discrete values also seems a little bit arbitrary.

The authors agree, there is some arbitrariness to the selection of discrete variables. We tested several different combinations of discrete variable selection, and included
the rules that worked the best for us. Although for specific cases there may be a better method, in general the rules we provide worked well (see the first paragraph of section 2.2). We have added the following paragraph to Section 2.2 that addresses this concern:

"The process outlined to select the discrete variables used in the parameterization is recommended as a starting point, and when computational resources or time is limited. We tested many different methods of how to determine the discrete values, but found that the method shown above consistently produced wind farm layouts with high energy production. With sufficient resources, some scenarios may benefit from optimizing with a different ratio of boundary turbines, or different initializations of the boundary grid. However, the results discussed in this paper were produced with the method given in this section. Because these variables are discrete, they cannot be included as design variables when using a gradient-based optimization method, because the function space would be discontinuous. But, a gradient-free optimization may benefit from including some of these discrete variables as design variables in the optimizations."

It is stated in lines 87- 88, the discrete values remain fixed, but then again, you have the situation that there are too many grid points inside the boundary (when dx and dy are small), if you have to put 45% turbine around the boundary, you will have to remove some grid points, then which ones to remove according to what rule?

With very small wind farms (much less 4 rotor diameter average turbine spacing), our suggested discrete variable initialization would not be able to meet spacing constraints and boundary constraints. The optimizer should be able to handle this, and adjust dy and dx such that all the constraints are satisfied, however it would be helpful to start with a feasible layout. We have added the following to section 2.2 to clarify this:

"For extremely small wind farms, with an average turbine spacing much less than 4 rotor diameters, it may be impossible to initialize the turbine rows with $dy$ equal to be

four times $dx$ and meet the minimum spacing constraints. In this case, the discrete row variable initialization would need to be adjusted."

For even more extreme cases, where you can't fit all of the turbines in the wind farm because the boundary is too small, you would just need to reduce the number of turbines desired in the wind farm and repeat our initialization process. This needs to be done in any layout optimization however, and is not unique to our study.

2. Some shortages in wind farm modelling.

First, in lines 117, it says "the turbulence intensity is equal to 0.0325", but shouldn't turbulence intensity change upon the wind speed?

Our revised paper will include results with the full 2016 Bastankhah Gaussian wake model rather than a simplified version. Details on this model will be included in the revised draft. This model also has a $k$ value that is dependent on the freestream turbulence intensity, which we will clarify.

Second, according to Eqs.(3-4), you use the wake deficit at the rotor center to represent the average wake deficit on the whole rotor, since there is no integration over the rotor area in Eq. (4). This is problematic, as the profile of wake deficit is a Gaussian shape, and the one point deficit in the rotor center could be overestimating the mean deficit, if the two turbines are perfectly aligned.

Our revised paper will include results where several wind speeds are sampled across the wake and averaged to find the effective wind speed used in the power calculation. This dramatically increases computational expense, but reduces the possibility of overestimating the mean deficit from the Gaussian wake.

Third, there are only 5 wind speeds, and 23 wind direction sectors used in the wind resource modelling, according to Eq. (7). It has been shown in some studies that you need finer discretization, for example in (Feng and Shen 2015) in your references. This kind of coarse discretization could give you artificially optimistic AEP gains. You may also check the follow paper for recommended discretization:

Feng, Ju, and Wen Shen. "Modelling wind for wind farm layout optimization using joint distribution of wind speed and wind direction." Energies 8, no. 4 (2015): 3075-3092.

The revised draft will report the optimized AEP calculated with 360 wind direction bins and 50 wind speed bins. To avoid restrictive computation time, the optimizations are still run with fewer bins, but the final results will be reported with finer discretization.

3. The missing of comparison to gradient-free optimization technique.

I understand the focus of this study is on the proposed parameterization. But without direct comparison of the gradient based optimizer to some gradient free ones, e.g., GA or RS, it looks unfounded and somehow biased for a lot of claims that says the gradient free method will be infeasible, or perform worse.

Both gradient-based and gradient-free methods improve.ÂăÂăWe aren't claiming gradient-free is worse than gradient-based at the smaller dimension.ÂăÂăThe main motivation for this work is to make these kinds of problems tractable for gradient-free approaches.ÂăÂăIt is well documented that gradient-free methods don't scale well to large number of design variables. Here are just a few:

Lyu, Zhoujie, Zelu Xu, and J. R. R. A. Martins. "Benchmarking optimization algorithms for wing aerodynamic design optimization." Proceedings of the 8th International Conference on Computational Fluid Dynamics, Chengdu, Sichuan, China. Vol. 11. 2014.

Rios, L. M., and Sahinidis, N. V., "Derivative-Free Optimization: a Review of Algorithms and Comparison of Software Implementations," Journal of Global Optimization, Vol. 56, No. 3, 2013, pp. 1247–1293. doi:10.1007/s10898-012-9951-y, URL https://doi.org/10.1007/s10898-012-9951-y.

Zingg, David W., Marian Nemec, and Thomas H. Pulliam. "A comparative evaluation of genetic and gradient-based algorithms applied to aerodynamic optimization." European Journal of Computational Mechanics/Revue Européenne de Mécanique Numérique 17.1-2 (2008): 103-126.

Thomas, J. J., and Ning, A., "A Method for Reducing Multi-Modality in the Wind Farm Layout Optimization Problem," Journal of Physics: Conference Series, Vol. 1037, No. 042012, Milano, Italy, The Science of Making Torque from Wind, Jun. 2018. doi:10.1088/1742-6596/1037/4/042012

Ning, A., and Petch, D., "Integrated Design of Downwind Land-Based Wind Turbines Using Analytic Gradients," Wind Energy, Vol. 19, No. 12, pp. 2137–2152, Dec. 2016. doi:10.1002/we.1972

But with only 5 design variables both gradient-free and gradient-based methods should produce good results. We will add the above citations on the poor scaling of gradient-free optimization with few design varaibles.

Also do you have bounds on the design variables?

No. The only constraints were the boundary and spacing constraints mentioned in Section 4 of the paper. The following text has been added to the paper:

"No bound constraints, or additional constraints were used to define where the turbines must lie."

How are the constraints handled in the optimization process? Penalty function?

We used the optimizer SNOPT, which is an SQP algorithm. A sentence in Section 4 was modified to clarify this:

"We used the optimizer SNOPT, which is a gradient-based optimizer that uses sequential quadratic programming, and is well suited for large-scale nonlinear problems such as the wind farm layout optimization problem "

Below is a note referring to the documentation of SNOPT for further details:

https://web.stanford.edu/group/SOL/guides/sndoc7.pdf

4. The claim on the infeasibility of gradient-free technique for large wind farm is unfounded.

AS stated in lines 9-10, Our presented method unlocks the ability to optimize and study large wind farms, something that has been mostly infeasible in the past. But I found this unfounded, you can check the following paper:

Wagner, Markus, Kalyan Veeramachaneni, Frank Neumann, and Una-May O'Reilly. "Optimizing the layout of 1000 wind turbines." European Wind Energy Association Annual Event 205209 (2011).

This was reworded to say:

"Our presented method facilitates the study and both gradient-free and gradient-based optimization of large wind farms, something that has traditionally been less scalable with increasing numbers of design variables."

Also engineering wake models are very fast to run, it shouldn't become too heavy or

even infeasible for a gradient-free optimizer applied to a wind farm with 100 turbines, even if needs 10000 evaluations.

Excellent thought. We do make several claims throughout the paper about the infeasibility of wind farm layout optimization with increasing design variables, specifically in regards to gradient-free optimization. First let's look at the paper you mentioned above. In this paper, they optimize the layout of 1000 wind turbines, which is impressive. However, we see that they used 20 cores, and a single optimization still took 12 days. On a single core, they estimate that a single optimization would take about 140 days! Now, for most applications, we believe that 140 days is infeasible, or at the very least restrictive. Even 12 days limits the study of wind farms due to computation expense.

Now let's compare to our experience. Even with our fast engineering wake model, fewer turbines, and exact gradients, the direct optimizations for the first draft of our paper took 4-6 hours each to complete. With the updated wake model, (added ct curve, increased number of samples in the wake, finer bin samples to evaluate the final AEP values), the optimizations are taking at least 10 hours, some much longer. These additions really start to add up. This is with exact analytic gradients, so no additional function calls are happening to estimate gradients. Central-differenced gradients would take (at least) 3 times as long to optimize, and a gradient-free approach longer still. Additionally, we are using only one core in each optimization! Although a week or a month or longer to optimize a wind farm may not be restrictive if it is a one off occurrence, this is almost never the case. Usually the objective is to optimize the farm several times with different parameters and considerations, to see how the layout and performance is affected. Cases such as this benefit greatly fast optimization, which is provided by our presented parameterization.

Additionally, higher fidelity models are not very fast to run. In these cases, reducing the number of function calls required to optimize by several orders of magnitude or

more is very important. As computation improves, these higher fidelity models will be used in wind farm layout optimization. In these cases, efficient optimization will play an important role.

5. Some very relevant references are missing.

Especially studies on grid-like layout optimization. The parameterization for the inner grid has been proposed in a similar way in some studies already. You may find the following two of interest:

González, Javier Serrano, Ángel Luis Trigo García, Manuel Burgos Payán, Jesús Riquelme Santos, and Ángel Gaspar González Rodríguez. "Optimal wind-turbine micro-siting of offshore wind farms: A grid-like layout approach." Applied energy 200 (2017): 28-38.

Neubert, A., A. Shah, and W. Schlez. "Maximum yield from symmetrical wind farm layouts." In Proceedings of DEWEK. 2010.

We added a citation for the paper by Neubert, Shah, and Schlez. The paper by González et al. was already cited on line 41.

Some minor issues:

1. It is stated in lines 20-25 that "Although these methods can be highly effective for small numbers of design variables, the computational expense required to converge scales poorly, approximately quadratically, with increasing numbers of variables. Because of this poor computational scaling, many companies and researchers have been limited in the size of wind farms they can optimize, as the number of variables typically increases with the number of turbines." But I doubt that's the case, since there are already large wind farms be designed and built in the world. Also optimization studies

have been conducted for large wind farms, such as Horns Rev 1 with 80 turbines, as in one of your references (Feng and Shen, 2015).

Refer to our discussion to your statment: "Also engineering wake models are very fast to run, it shouldn't become too heavy or even infeasible for a gradient-free optimizer applied to a wind farm with 100 turbines, even if needs 10000 evaluations."

Yes it can and has been done. But it has been at great computational cost. Our presented parameterization makes these types of studies much more manageable.

2. Lines 31-32 "Power losses of 10–20% are typical from turbine interactions within a wind farm (Barthelmie et al., 2007, 2009; Briggs, 2013), and can be as high as 30–40% for farms with closely spaced wind turbines (Stanley et al., 2019)." This is somehow misleading, power losses of 30-40% are the worst wake case, which doesn't happen that frequent in reality. So the actually AEP loss due to wake effects should be usually lower than 10-20%.

This was reworded for clarification:

"Power losses of 10–20% are typical from turbine interactions within a wind farm, and can be as high as 30–40% for farms with turbines spaced within 3 rotor diameters of each other."

These values don't refer to worst case, but are in fact the overall wake loss (refer to the cited paper for more details).

3. Rosenbrock function is used to demonstrate the convergence of gradient based optimizer scales better than gradient-free methods. First, you need to show what is Rosenbrock function, or at least provide a reference.

A reference was provided.

Second, this function is a function that we actually know where are the optimums, thus, we can easily see when it has converged to a local minimum. But in real life applications, we often can't analytically prove that we have reached a local minimum, such as in layout optimization.

True, which is why the Rosenbrock function is a good test function for determining the efficiency of optimization algorithms. Figures 1 and 11 of the paper and the associated discussions demonstrate the multimodality and difficulty of the wind farm layout optimization problem.

Third, for such problem, converge faster (typically for gradient based methods) is just one aspect, the other aspect is the quality of the optimized results, i.e., whether the solution is close to the global optimum. Usually it is know that gradient free methods converges slower but has a higher probability to reach the global optimum, while gradient based methods converge faster but are also easier to be trapped in a local minimum.

Correct. However with large problems, convergence speed is a very important aspect. This simple example was used to highlight (and we feel that it is done so effectively) the huge importance of efficient computation, and the extreme effects that inefficient optimization can have on computation time.

4. Eq. (6), U_mean should be scale factor of the Weibull distribution. Note that the scale factor is not the same thing as the mean wind speed, instead the mean wind speed should be a function of scale factor and shape factor.

This will be corrected in the revised manuscript. Final results will include this correction.

5. Line 275-276 states that "BG parameterization, cabling requirements can be clearly minimized by running cables across each of the rows, and around the boundary without the need for complex cabling algorithms." This is not true, as you still need to decide the location of sub-station, the exact topology of the cables and select cable types for different connections, thus, not necessarily easier than any random layout.

This claim was removed from the paper.